A Cellular Potts Model of the interplay of synchronization and aggregation

Una Rose
Glimm Tilmann glimmt@wwu.edu
Department of Mathematics, Western Washington University , Bellingham , WA , United States of America
Gillespie Joseph
Electronic publication date: 2024 Feb 29
Publication date: 2024
Volume: 12
Electronic Location ID: e16974
Received 2023 Sep 6; Accepted 2024 Jan 29
Copyright: ©2024 Una and Glimm
Copyright year: 2024
Copyright holder: Una and Glimm
License: This is an open access article distributed under the terms of the Creative Commons Attribution License, which permits unrestricted use, distribution, reproduction and adaptation in any medium and for any purpose provided that it is properly attributed. For attribution, the original author(s), title, publication source (PeerJ) and either DOI or URL of the article must be cited.
License URL: https://creativecommons.org/licenses/by/4.0/

Keywords: Cellular Potts Model, Synchronization, Aggregation, Biological clocks, Mathematical modeling

Funding: The John Templeton Foundation #62220 A Jarvis Memorial Summer Research Stipend from Western Washington University Tilmann Glimm was supported by the John Templeton Foundation (#62220). The opinions expressed in this paper are those of the authors and not those of the John Templeton Foundation. Rose Una was supported by a Jarvis Memorial Summer Research Stipend from Western Washington University. The funders had no role in study design, data collection and analysis, decision to publish, or preparation of the manuscript.

==============================
We investigate the behavior of systems of cells with intracellular molecular oscillators (“clocks”) where cell-cell adhesion is mediated by differences in clock phase between neighbors. This is motivated by phenomena in developmental biology and in aggregative multicellularity of unicellular organisms. In such systems, aggregation co-occurs with clock synchronization. To account for the effects of spatially extended cells, we use the Cellular Potts Model (CPM), a lattice agent-based model. We find four distinct possible phases: global synchronization, local synchronization, incoherence, and anti-synchronization (checkerboard patterns). We characterize these phases via order parameters. In the case of global synchrony, the speed of synchronization depends on the adhesive effects of the clocks. Synchronization happens fastest when cells in opposite phases adhere the strongest (“opposites attract”). When cells of the same clock phase adhere the strongest (“like attracts like”), synchronization is slower. Surprisingly, the slowest synchronization happens in the diffusive mixing case, where cell-cell adhesion is independent of clock phase. We briefly discuss potential applications of the model, such as pattern formation in the auditory sensory epithelium.

Introduction

Synchronization of coupled oscillators is a common phenomenon in nature, for instance the emergence of synchrony in neural networks (Singer, 1999; Hansel, Mato & Meunier, 1995; Uhlhaas et al., 2009), the synchronization of fireflies flashing (f.ex. Faust, 2010; Ramírez-Ávila et al., 2019; Sokol, 2022), the coordination of circadian rhythms in eusocial colonies (Siehler, Wang & Bloch, 2021; Frisch & Koeniger, 1994) or the synchronization of intracellular molecular oscillatory processes in developmental biology (Jiang et al., 2000; Bhat et al., 2019; Venzin & Oates, 2020; Deneke et al., 2016). Beginning with the seminal works of Winfree (e.g., Winfree, 1987) and Kuramoto (e.g., Kuramoto, 1984) in the 1970s, mathematical models of synchronization phenomena in networks of coupled oscillators have been an intense area of study; see e.g., the books by Pikovsky, Rosenblum & Kurths (2003) and Boccaletti et al. (2018), or surveys by Dörfler & Bullo (2014) and Rodrigues et al. (2016).

The interplay of synchronization and spatial aggregation in systems of moving interacting oscillators is much less well studied in models. The most influential work in this direction is the investigation of swarming oscillators (“swarmalators”) by OKeeffe, Hong & Strogatz (2017) and subsequent works by Sar et al. (2022), OKeeffe, Ceron & Petersen (2022) and Barciś & Bettstetter (2020). These “swarmalators” are mass points with internal oscillators (“clocks”) which attract or repel each other according to their clock phase difference, and also interact with each other via Kuramoto-like interactions.

One important potential application of systems of interacting moving oscillators is the aggregation of biological cells mediated by cell–cell adhesion in aggregative multicellularity, e.g., the slime bacteria myxobacteria (Shimkets & Kaiser, 1982; Zusman et al., 2007; Peruani et al., 2012; Thutupalli et al., 2015), or the slime mold Dictyostelium discoideum (Bonner, 2008; Van Oss et al., 1996; Marée & Hogeweg, 2001). Indeed, these two systems also exhibit intracellular oscillators (Gregor et al., 2010; Alber, Jiang & Kiskowski, 2004; Guzzo et al., 2018; Arias Del Angel et al., 2020).

Another application is in vertebrate embryogenesis, where members of the Hes gene family, which is known to play a central roles in determining cell fate in development, display sustained intracellular oscillations in expression patterns (Kageyama, Ohtsuka & Kobayashi, 2007; Hirata et al., 2002). Synchronized Hes oscillations are crucial for somitogenesis (recently reviewed by Carraco, Martins-Jesus & Andrade (2022)), providing the experimental foundation for the famed clock and wavefront model of somite pattern formation originally proposed by Cooke & Zeeman (1976), see also Murray, Maini & Baker (2011). Another example of the interplay of spatial pattern formation and intracellular oscillators is in vertebrate limb development, where Hes1 was found to play a role in regularizing the spatial pattern of precartilage condensations, aggregates of mesenchymal cells (Bhat et al., 2019; Newman, Bhat & Glimm, 2021). While there is no known direct link between the Hes family and cell–cell adhesion, there is a well-established link between Notch signaling and Hes expression Kageyama, Shimojo & Isomura (2018), Kageyama, Ohtsuka & Kobayashi (2007). The Notch pathway controls cell communication between neighboring cells, but also may directly mediate cell–cell adhesion (Murata & Hayashi, 2016), making assumed mutual interdependence between intracellular oscillations and cell–cell adhesion plausible.

Motivated by this, Glimm & Gruszka (2024) recently suggested a partial differential equations (PDE) model that takes into account local cell–cell adhesion (as opposed to attraction independent of the distance of two cells as in O’Keefe et al.’s swarmalators), as well as a local Kuramoto-like interaction of oscillators. In a linear stability analysis, they identfied a number of emerging behaviors –the possibility of aggregation or dispersion can be combined with either global synchronization, only local synchronization (synchronized patches), or incoherence.

A PDE model has the advantage of being amenable to analytic methods, but has the drawback of dealing with cell densities instead of modeling cells individually. This means that the effects of variations of cell size and shape are not modeled. To address this, we present a model based in part on the work of OKeeffe, Hong & Strogatz (2017) and Glimm & Gruszka (2024), but where cells are modeled via the Cellular Potts Model (CPM) (Graner & Glazier, 1992). The CPM, also known as the Glazier-Graner-Hogeweg model, is a computational lattice-based model that allows for spatially extended cells and incorporates fluctuations in cell size and cell shape.

Our minimalistic model incorporates cells with an intracellular oscillator. We subsume the complex dynamics of intracellular oscillations into a single variable and assume that adhesion between two cells depends on the relative oscillator phases, and that adjacent cells influence each others’ oscillators. (Portions of this text were previously published as part of a thesis; see Una, 2023). More specifically, in our CPM framework, each cell is associated with an internal oscillator, or cell clock. Clocks interact locally via Kuramoto-like interactions, where the type and strength of interactions is encapsulated by a parameter K. Cell–cell adhesion between neighboring cells is influenced by the clock phase differences with three different qualitative possibilites encoded by a parameter J: “like attracts like”, where cells in the same phase adhere to each other maximally (J > 0); “opposites attract”, where cells in opposite phases adhere to each other maximally (J < 0); and the purely diffusive case where cell–cell adhesion is independent of clocks (J = 0). We investigate the effects of the parameters J and K on oscillator synchronization and spatial aggregation. We find four distinct types of steady phase states emerging from random initial conditions: global synchronization, local synchronization, incoherence, and anti-synchronization (checkerboard pattern). We define the four distinct phase states qualitatively with a phase diagram and quantitatively with order parameters. Of particular interest is the case of global synchronization, which occurs when the Kuramoto interaction parameter K is positive, i.e., neighboring cells seek to synchronize their clocks. Here we find that initially, synchronization advances fastest for negative J and slowest for positive J. But in the long run, remarkably, synchronization is slowest in the “border case” J = 0, i.e., the purely diffusive case when cell–cell adhesion is independent of clock phase. We give an intuitive explanation of this phenomenon.

Our work is a high-level generic model of the interplay of intracellular oscillations and cell–cell adhesion. It is motivated by examples from development biology and aggregative multicellularity, but is not a model of a specific experiment. Instead, we seek to provide an investigation of the broad types of patterns in space and synchronizations that are possible with a set of minimal assumptions. However, we discuss potential applications to concrete findings about pattern formation in the auditory sensory epithelia of many species, where checkerboard patterns composed of sensory hair and supporting cells are established in development and crucial for proper functioning (Togashi et al., 2011; Katsunuma et al., 2022).

Methods

The model

The model uses the framework of the Cellular Potts Model (CPM; see Graner & Glazier (1992)) based on a rectangular lattice. Each cell is modeled as a group of lattice sites and is associated with an internal oscillator (“clock”). Adhesion between cells is incorporated into the model via the Hamiltonian, an energy function. Time evolution is modeled via an energy minimization procedure in space and an updating rule for the clocks. (See Fig. 1 for a sketch of the schematics of the CPM.)

Figure 1 Schematics of the cellular potts model.

Left: Cells are modeled as sets of lattice sites. We picture three cells with indices (“spins”) 1–3. The index 0 denotes the extracellulcar matrix (medium). Each cell has a clock whose current value is represented by the color of the cell. Each configuration of the lattice has a Hamiltonian (energy) H, which encodes various biological phenomena such as adhesion and elasticity. Right: Time evolution is modeled via so-called “spin flips”. A lattice site is chosen at random and a change of its index to that of a neighboring cell is proposed. This change is accepted with the displayed probability which depends on the change ΔH in the Hamiltonian it causes. Here T is a given constant, the “temperature.” See the text for more details.

More specifically, lattice sites are denoted by bold variables i = (i1, i2) or j = (j1, j2). At every time step, each lattice site i belongs to one of N cells or to the extracellular matrix (ECM). Cells are numbered with an index that ranges from 1 to N. We use the notation σt(i) = 0 if the site i belongs to the medium at time step t, and σt(i) = s if i belongs to the sth cell (s = 1, …, N). Additionally, each cell s = 1, …, N has an internal clock. This clock is a time-dependent scalar θst. For notational simplicity, we will suppress the superscript t unless necessary.

The governing Hamiltonian is (1) H= ∑neighboringlattice sitesi,j1−δσi,σjfσi,σj+λ∑cellsAreas−Atarget2.

The Hamiltonian encodes the compressibility of the cells and cell–cell adhesion. The second term of the Hamiltonian is a cell area constraint term. Here A(s) is the area of the cell with index s, i.e., the number of lattice sites it occupies. Atarget is the target area, a fixed reference area, and λ is a parameter that encodes the compressibility of the cell: the larger λ is, the less fluctuations in size there are. (Our choices of Atarget and λ are based on the work of Zhang et al. (2011); see Table 1.)

Table 1 Parameter table.

An asterisk * marks parameter values taken from Zhang et al. (2011).

Parameter	Value	Description	
Lattice dimensions	126 × 126	Cartesian (square) lattice size with periodic boundary conditions	
T	20	Temperature*	
θ 0	U([0, 2π])	Uniform initial distribution of initial clock phases	
A target	25	Target cell area*	
λ	25	Area constraint coefficient*	
ω	0.001	Clock speed	
J MM	0	Medium–medium contact local product adhesion energy*	
JCM = J0	16	Medium-cell contact local product adhesion energy*	
	2	Neighbor order*	
N	445	Cell count	

Crucially, the first term of the Hamiltonian describes cell–cell adhesion between adjacent cells. The symbol δij is the Kronecker delta: δi,j = 1 if i = j, and δi,j = 0 if i ≠ j. The cell–cell adhesion term f(σ1, σ2) depends on the clock values of the adjacent lattice sites with indices σ1 and σ2 and is given by (2) fσ1,σ2=J01−Jcosθσ1−θσ2ifσ1≠0andσ2≠0J0ifσ1=0orσ2=0

Recall that the index σ = 0 represents extracellular matrix, which has a contact energy of J0. The parameter J encodes the effect of clock phases on adhesion. For J > 0, the function f(σ1, σ2) is minimized when θσ1 = θσ2, so cells of like clock phases adhere to each other the strongest (“like attracts like”). For J < 0 in contrast, f(σ1, σ2) is minimized when θσ1 = θσ2 + π, and so then cells of opposite clock phases adhere to each other strongest (“opposites attract”).

Time evolution is modeled via so-called spin flips. For this, a lattice site is selected randomly and it is proposed to change its index to that of a neighboring lattice site. Such a spin flip is accepted with a probability that depends on the change in the Hamiltonian function ΔH it would entail. Specifically, it is given by (3) Prob(spin flip)=1ifΔH≤0e−ΔH/TifΔH>0

where T is the so-called temperature of the system. Higher temperatures make it more likely for spin flips to occur but also make it more likely for cells to fragment, dissolving into each other. A Monte Carlo Step (MCS) consists of the number of spin flips corresponding to the total number of sites in the lattice. Simulation time is commonly measured in MCS.

Each cell s = 1, …, N in the model has an internal oscillating clock. For each cell s = 1, …, N, we update its cell clock θst each MCS via (4) θst+1=θst+ω⋅1+K⋅1# neighbors ofs∑neighboruofs sinθu−θs

Note that a cell’s internal clock advances at the uniform clock speed ω, but is additionally influenced by the clocks of its neighbors in a Kuramoto-type way (Kuramoto, 1984). Here K is the clock coupling strength between neighboring cells. It controls how neighboring cells’ clocks influence each other. For K > 0, neighboring cells seek to synchronize their clock phases. For K < 0, neighboring cells seek to anti-synchronize their clocks.(The meanings of the crucial parameters J and K are summarized in Table 2.) As in other models (OKeeffe, Hong & Strogatz, 2017; Glimm & Gruszka, 2024), we note that all equations only depend on differences of clock phases, so that they are invariant under shifts in clock phase space in the form θst→θst−ω⋅t. Effectively, this means we can assume that the uniform clock speed is zero and that only the terms with the K −factor in the update Eq. (4) enter into changes of the clocks at each MCS. In this sense, the choice of ω actually does not enter into the calculation apart from scaling K.

Table 2 Meaning of parameters J and K.

Parameter	Sign	Effect	
J	J > 0	Adherence strongest for same clock phase (“Like attracts like”)	
	J < 0	Adherence strongest for opposite clock phases (“Opposites attract”)	
K	K > 0	“Neighbors seek to synchronize”	
	K < 0	“Neighbors seek to anti-synchronize”	

Parameters

The aim of our model is a high-level generic investigation of the interplay of adhesion and synchronization with minimal assumptions. Since it is not modeled on a concrete experimental setup, we chose to adopt the approach of Zhang et al. (2011), who used CPM simulations to investigate and validate Steinberg’s differential adhesion hypthesis (Steinberg & Takeichi, 1994). The parameter values Zhang et al. (2011) chose are based on experimental studies by Armstrong (1971) and Steinberg & Takeichi (1994). Armstrong used retinal cells from chicken embryos, Steinberg used mouse L-cells. The paraments of Zhang et al. (2011) were not the result of direct measurements, but calibrated via matching quantities such as cell size and velocity distributions of individual cells to retinal cell data (Mombach et al., 1995; Mombach & Glazier, 1996) and ensuring that cells do not fragment. Our parameter values are indicated in Table 1. The lattice length scale is approximately 2 µm per pixel, the time scale is about 10,000 MCS per hour (Zhang et al., 2011). Note that our confluency is 70%, a typical value for in vitro experiments. Our oscillator velocity ω = 0.001(clock change per MCS) corresponds to a period of about 7,000 MCS and thus very roughly on the order of one hour. This is the same order of magnitude as oscillations of Hes1 expression in somitogenesis (Kageyama, Ohtsuka & Kobayashi, 2007). We point out that in our model, only the differences between clock phases of neighboring cells matters for dynamical updates. Since all oscillators are assumed to have the same clock velocity ω, this means that simulations are actually independent of the value of ω and only depend on the effective value of the clock coordination parameter K. Accordingly, in our graphs, the color of the cells correponds to the phase shift relative to a clock that advances at a steady speed ω. We acknowledge that this independence of our simulation results on ω breaks down if we drop the assumption that every cell has the exact same speed in favor of a more realistic distribution of speeds, but this is outside the scope of the current work.

Order parameters for characterizing phases

In the Results section, we will show that our model displays qualitative different types of behaviors for different choices of the parameters J and K. This is analogous to phase states in statistical physics. Phase transitions are often described quantitatively by order parameters, approppriately defined quantities that characterize the state of the system; see e.g., Nolting (2018). We define here three order parameters for our model. In the Results section, we will use these to characterize the phases.

The first order parameter is a classical parameter quantifying synchronization due to Kuramoto (1984). It is given by (5) rglobal=1N∑j=1Neiθj

where N is the total number of cells in the simulation. The resulting rglobal then satisfies 0 ≤ rglobal ≤ 1. The parameter rglobal quantifies how synchronized all cells’ clocks are at a specified MCS. If rglobal = 1, all the cells’ clocks are synchronized. If rglobal = 0, all the cells’ clocks are entirely unsychronized or anti-synchronized, meaning that all of their values are spread around the time clock evenly such that all values cancel each other out or there are synchronized groups of opposite phases that cancel each other out.

The second order parameter is a local version of rglobal. For each cell, we compute a modified r −value via an average over its nearest neighbors (order 2). These values are then averaged over all cells: (6) rlocal=1N∑cellk1sk ∑neighborjofkeiθj

where the first sum is taken over all cells in the simulation, sk is the number of neighbors (order 2) of cell k and the second sum is taken over all neighbors of cell k. Note that rglobal = 1 (complete global synchronization) implies rlocal = 1, but not vice versa. Also note that complete randomness of phases does not typically lead to rlocal = 0 because the second sum in the Eq. (6) is taken over a relatively small number, leading to large random variations. Numerically, we found that incoherence corresponds to a value of rlocal of roughly 0.2; see also the discussion in the subsection “Phase Diagram”.

The third order parameter (“checkerboard parameter”) quantifies the extent to which neighboring cells’ clock phases are in opposite phase (phase difference π). We need some tolerance since neighboring cells will not be completely in opposite phase all the time. We chose a tolerance of 6.25% from total anti-synchronization between neighboring cells. More explicitly, ψ=total number of pairs of neighboring lattice sitesk,jsuch thatθk−θj∈π−π8,π+π8(modulo2π)

Computational implementation

To implement the Cellular Potts Model, we used the open-source software CompuCell3D, version 4.2.5 (Swat et al., 2012) on a standard PCs (Intel(R) Core(TM) i5-6600 CPU @ 3.30 GHz, 16 GB RAM). Further analysis of the simulation data and generation of graphics were performed with Matlab. Figure 2 displays the results of 49 simulation runs, each of which took roughly 8–10 h individually.

Figure 2 Parameter sweep of the model showing the state after 250,000 MCS.

Cells are colored according to their clock phase. (Specifically, the colors show the clock phase difference relative to a standard clock moving at constant clock speed ω starting at 0.) Values shown are J ∈ { − 0.95,  − 0.6333,  − 0.3167, 0, 0.3167, 0.6333, 0.95} and K ∈ { − 1,  − 0.6667,  − 0.3333, 0, 0.3333, 0.6667, 1}. Simulation runs were each of N = 445 cells and identical initial conditions (random distribution of clock phases).

Results

Phase diagram

We concentrate on investigating the effects of the two parameters J and K encoding the interactions of clock synchronization and cell–cell adhesion. See Table 2 for the physical meaning of these parameters. The ranges −1 < K < 1 and −1 < J < 1 are meaningful for simulations1. Starting with random initial conditions for cell positions and clock values, we ran simulations to determine the long-term behavior for different values of J and K. The results are summarized in the phase diagram in Fig. 2. We used movies of the simulations to confirm that the simulation duration of 250,000 MCS was sufficiently long to guarantee convergence to an obvious phase state; see also Section “Dynamics of Synchronization”. (Movies of all simulations are available online on the Zenodo repository at DOI 10.5281/zenodo.10681751).

The phase diagram in Fig. 2 clearly displays qualitatively different types of behaviors. To investigate these phases, we use the three order parameters defined in Order Section “Parameters for Characterizing Phases”. Their heat maps are shown in Fig. 3. Note that all except rlocal display very sharp step-like transitions between small and large values. This allows to quantify the different phases. We identify four different phases: The first is characterized by large rglobal. We call it “global synchronization.” Large values of the checkerboard parameter ψ characterize another phase, which we call “anti-synchronization.” (Specifically, we can consider ψ > 1000.) There are two more phases, which we call “incoherence” and “local synchronization”. The transition between incoherence and local synchronization is more gradual as clusters of synchronized cells become smaller with decreasing negative K. Nevertheless, we can utilize the parameter rlocal to distinguish between definite local synchronization (rlocal ≈ 1) and definite incoherence (rlocal ≈ 0.2).

Figure 3 Heat maps of the three order parameters (rglobal, rlocal, checkerboard parameter ψ).

Note the jump-like transitions along the lines K = 0 for rglobal, as well as J = 0 and K = 0 for ψ. There is a similar sharp sharp transition of rlocal along K = 0 for J < 0 and a curve in the quadrant J > 0, K ≤ 0. This separates the parameter space into four phases: global synchronization (rglobal ≈ 1); local synchronization (rlocal ≈ 1, rglobal = 0); antisynchronization (checkerboard pattern) (large ψ); and incoherence (all order parameters “small”, see Fig. 4).

Figure 4 summarizes the phases. Global synchronization in the right half K > 02 and anti-synchronization (checkerboard pattern) in the quadrant J < 0, K ≤ 0 are both readily visibly identifiable. This is straightforward to understand: For K > 0, neighboring cells seek to synchronize; and since all cells adhere to each other, this eventually leads to global synchronization for all values of J. (The dynamics of synchronization differ by whether J is positive or negative though; we investigate this in “Dynamics of Synchronization”.) The anti-synchronized phase state is characterized by cells of opposite phases attracting each other (J < 0). Furthermore, cells seek to anti-synchronize with their neighbors (for K < 0). The resulting behavior is a checkerboard-style distribution of cell phases where cells minimize their energy by surrounding themselves with cells of the opposite phase.

Figure 4 Diagram of the four phases of the model.

See Fig. 3 for order parameters.

The other two phases (local synchronization (synchronized spatial clusters) and incoherence (quasi-random spatial distribution of phases)) both occur in the quadrant J ≥ 0, K ≤ 0. For very negative K and small J, we have incoherence. In the case large J and K close to 0, cells tend to sort by phases and clusters de-synchronize sufficiently slowly that persistent locally synchronized clusters form.

Movies show that global synchronization and anti-synchronization are essentially static distributions, where the cells’ positions and shapes fluctuate, but the overall spatial distribution of clock phases stays the same. Incoherence and local synchronization are dynamic phases, where cells or cell clusters in different clock phases constantly move relative to each other, yielding a behavior in which each snapshot in time is different, but characteristic of the typical distribution.

To further characterize the incoherence phase, we compared the resulting distributions of clock values to the distribution obtained by chance. The results are summarized in Fig. 5. Note that the distributions of clock values for the incoherent phase is essentially indistinguishable to a uniform random distribution on the interval [0, 2π). There is a marked difference to the anti-synchronization (checkerboard) phase, with two peaks at the distance π. Interestingly, the “local synchronization” case also gives a distribution indistinguishable from chance. This is because in the representation of Fig. 5, all spatial information is lost, so spatially completely mixed clock phases and spatially clustered clock phases give similar distributions.

Figure 5 Histograms of distributions of clock phases for some of the simulations from Fig. 2.

Each of the N = 445 cells gives one data point in the interval [0, 2π). We used 20 bins. Here “random” denotes a random sample obtained by choosing n = 445 random numbers in [0, 2π) with uniform probability distribution. Note that the resulting distributions for the “incoherent” phase data is essentially indistinguishable from the random distribution. Interestingly, the same is true for the example of the “locally synchronized” phase, but not the “anti-synchronized” (or checkerboard) phase, which is bimodal.

Figure 6 Progress of synchronization for K = 1 and different values of J.

Cells are colored by the clock phase difference relative to a standard clock moving at constant clock speed ω as in Fig. 2.

Dynamics of synchronization

For positive values of K, cells’ clocks eventually globally synchronize as shown in Fig. 2. When we investigated the dynamical paths to synchrony however, we found substantial qualitative differences depending on the sign of the parameter J (positive, negative or zero). Figure 6 illustrates this with K = 1 and three different values of J. To quantify the initial progress of synchronization for different values of J, Fig. 7A shows the order parameter rglobal over time (MCS) up to 20,000 MCS. There is a clear hierarchy of J-values –the speed of synchronization is highest for the most negative value of J and decreases with increasing J with J =  + 0.95 corresponding to the slowest rate of progress. This can be seen also by the first two columns in Fig. 6. Why is this? Again, Fig. 6 provides an important insight: For negative J, “opposites attract” and hence at t = 10, 000 MCS, it is clearly visible that for J =  − 0.95, there are large regions of synchronized cells with cells of the opposite phase interspersed. (For example regions of orange cells with interspersed small blue cells, some slightly fragmented.) This proximity of cells of opposite clock phases early on leads to rapid synchronization. In contrast, this is not the case for J =  + 0.6333 (positive J, so “like attracts like”) or J = 0 (clock phase has no influence on adhesion). One also observes that synchronized regions have more gradual transitions for J = 0 than for J =  + 0.6333. For instance for J =  + 0.6333, red and green cells tend to be separated by much smaller buffers of yellow and orange cells than is the case for J = 0.

Figure 7 Synchronization of cells over time for different values of J and fixed K = 1.

(A) Parameter rglobal as a function of time (MCS) up to t = 20, 000 MCS, covering the initial phase of synchronization. (B) Parameter rglobal as a function of time (MCS) up to t = 250, 000 MCS, covering the long term behavior. (C) Parameter rlocal as a function of time (MCS) up to t = 20, 000 MCS, covering the initial phase of synchronization. (All graphs are based on n = 10 runs for each curve.).

These observations can be quantified via the local parameter rlocal, shown in Fig. 7C: Here, the order is reversed and initially, the smaller J, the slower the rate of local synchronization. For negative J (“opposites attract”), the local synchronization even decays before increasing again.

The long-term rate of synchronization is plotted in Fig. 7B. The relationship between order of J −values and speed of synchronization seen in Fig. 7A is not preserved in the long run. Some of this may be due to the small sample size (n = 10). However, it is very clear that synchronization progresses most slowly for J = 0, the case where clock phase does not influence adhesion. At 250,000 MCS, complete synchronization had not been reached. (Illustrated also in Fig. 6 and Fig. 4.) This is in contrast to positive or negative values of J, where complete synchronization was achieved at that time. (Even for J = 0, synchronization was eventually achieved if the simulation was allowed to keep running; see time t = 400, 000 MCS in Fig. 6.)

Why is synchronization especially slow for J = 0? An intuitive explanation is as follows: Synchronization proceeds especially fast (for positive K) if the clock phase distribution has sharp spatial gradients, i.e., if many cells of very different clock phases are neighbors. In contrast, configurations with shallow clock gradients (gradual transitions of clock values) exhibit slower synchronization. With this principle, it is clear that for negative J, “opposites attract” and one indeed gets fields of cells with sharp clock gradients which synchronize quickly, as we noted before. But also the case of positive J (“like attracts like”) creates sharp gradients, but with a different mechanism: Early on, synchronized clusters form as cells with similar phases effectively move towards each other. Because of cell–cell adhesion, these clusters have relatively sharply defined edges (Fig. 6), meaning sharp clock gradients, again leading to faster synchronization. In contrast, for J = 0, clock gradients are much more gradual, allowing regions of different clock phases to persist longer.

Discussion

We proposed a model of synchronization and aggregation of individual oscillators. In contrast to previous models (OKeeffe, Hong & Strogatz, 2017; Glimm & Gruszka, 2024), the oscillators were not just point particles, but entities with extended, fluctuating boundaries motivated by the behavior of biological cells in vitro (Zhang et al., 2011). We found four distinct phase states on which a simulation run can settle into depending on the parameters J and K values. By changing whether or not cells seek to synchronize or anti-synchronize with their neighbors (K) and whether or not cells seek out others with the same or opposite phase (J), we found cells globally synchronize, only locally synchronize, globally anti-synchronize, or remained incoherent. The anti-synchronization phase (“checkerboard pattern”) is not found in the previous models with point particles and thus is made possible because of spatially extended cells.

For K > 0, we observed eventual global synchronization. The dynamics of synchronization differ by the parameter J though. Synchronization happens fastest for J < 0, the case where cells of opposite clock phases adhere most strongly. Indeed, there is a short transitory checkerboard pattern that quickly gives rise to uniform synchrony. This does not happen for J ≥ 0. Most surprisingly though is the result that synchronization happens the slowest in the case of J = 0, when mixing of cells is purely diffusive. In this case, shallow gradients of clock phases appear which persist for much longer than the sharper gradients for J > 0 or J < 0.

The possibility of persistent checkerboard patterns as one of the phases in our model is particularly interesting, since such checkerboard patterns are found in vivo in the auditory sensory epithelium of the cochlea of many species, composed of sensory hair cells and supporting cells (Togashi et al., 2011; Katsunuma et al., 2022). Both hair cells and supporting cells differentiate from pluripotent ectodermal cells (Wan, Corfas & Stone, 2013). The spatiotemporal determination of cell fate is arguably not completely understood, but it is influenced by members of the Hes/Hey genes, such as Hes1, which is known to undergo oscillatory expression in many developmental processes (Tateya et al., 2011; Hirata et al., 2002). Hes1 is thought to inhibit differentiation into hair cells. When Hes1 is knocked out, the checkerboard pattern is disturbed and in fact the number of hair cells is increased relative to the number of supporting cells (Tateya et al., 2011). Crucial for normal development is the expression of two adhesion molecules, nectin-1 and -3, which are produced by the hair cells and supporting cells. There is good evidence that heterotypic adhesion between these two cell types is the mechanism by which the checkerboard pattern is maintained. When nectin-3 was knocked out, the checkerboard pattern was disrupted, but there were still equal numbers of hair cells and supporting cells (Togashi et al., 2011).

Our model then gives a possible mechanism for the interplay of cell fate and spatial arrangements from an initially uniform field of pluripoptent cells, namely via intracellular oscillators such as Hes1 and their interplay with cell adhesion molecules. In this scenario, cells in a clock phase of high Hes1 expression would differentiate into supporting cells, those in a phase with low Hes1 expression into hair cells. Normal development then corresponds to the case of negative K(cells with high Hes1 suppress Hes1 production in neighboring cells) and negative J, meaning heterotypic adhesion. This yields a checkerboard pattern; see Fig. 2. The experiment of knocking out nectins by Togashi et al. (2011) then correspod to changing J from negative values to J = 0. This results in breaking up the checkerboard pattern into incoherent patterns, i.e., patterns without a correlation between clock phase and spatial position. Note that in these patterns, clock phases are spatially mixed and there is no bias towards any clock phase (Fig. 5), which is consistent with the observation that the numbers of hair cells and supporting cells remained equal (Togashi et al., 2011).

These observations point to the possibility of interesting insights our simple model, or more elaborate variants of it, can provide. Still, we need to stress that the model is minimalistic in that we concentrate on two core interactions –synchronization and aggregation. The cost of this generality is specificity. For further work, tayloring the core model to specific phenomea such as pattern formation in the auditory sensory epithelium as sketched above requires refining and enhancing it by matching parameters, but also potentially including phenomena that are not part of our current generic model such as cell polarity, chemotaxis, or dealing explicitly with cell differentiation.

Supplemental Information

Supplemental Information 1 Sample CompuCell3D code

Additional Information and Declarations

Competing Interests

Author Contributions

Data Availability

1 For K ≥ 1 or K ≤  − 1, individual clocks can stop or run backwards; see Eq. (4). Values of J with J > 1 or J <  − 1 mean nonpositive cell–cell interaction energies. This means that lattice sites corresponding to the same cell can have higher interaction energies than sites corresponding to different cells. This causes fragmentation of cells, an unphysical behavior.

2 Note that not all images in Fig. 2 for K > 0 show perfect global synchronization, but we verified that all approached synchronization eventually if simulations are allowed to continue; see also Fig. 6.

The authors declare there are no competing interests.

Rose Una conceived and designed the experiments, performed the experiments, analyzed the data, prepared figures and/or tables, authored or reviewed drafts of the article, code development and implementation, and approved the final draft.

Tilmann Glimm conceived and designed the experiments, performed the experiments, analyzed the data, prepared figures and/or tables, authored or reviewed drafts of the article, code development and implementation, and approved the final draft.

The following information was supplied regarding data availability:

The CompuCell3d code for two specific parameter values for K and J is available in the Supplemental File. The code used for other values was equivalent.

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
