# Peer review of "A Cellular Potts Model of the interplay of synchronization and aggregation"

_PeerJ, doi:10.7717/peerj.16974_

## Round 0.1 · original submission · Major Revisions

Dear Drs. Una and Glimm:

Thanks for submitting your manuscript to PeerJ. I have now received two independent reviews of your work, and as you will see, the reviewers raised some concerns about the research. Despite this, these reviewers are optimistic about your work and the potential impact it will have on research studying modelling synchronization and aggregation. Thus, I encourage you to revise your manuscript, accordingly, considering all of the concerns raised by both reviewers.

Please fully explain your rationale for model parameter choice.

Please edit the manuscript for Grammar and typos.

Ensure that your findings are thoroughly placed within the existing body of literature.

There are many suggestions by the reviewers that should greatly improve your manuscript.

Thus, I encourage you to revise your manuscript, accordingly, considering all the concerns raised by the three reviewers.

Good luck with your revision,

-joe

Reviewer 1 ·

Basic reporting

This paper presents a Cellular Potts model investigating the interplay between synchronization of oscillators and spatial aggregation of cells. The paper is overall well executed, with some areas needing additional depth and rigor. While I do not have the specialized mathematical background to critically assess the modeling equations and analysis, the results are interesting from a biological standpoint. The simulations reveal four distinct steady phase states: global synchronization, local synchronization, incoherence, and anti-synchronization, based on the variation of two parameters (J, and K).

Overall, while I find the study and the combination of mathematical models relevant, I think the paper needs revisions on the following topics:

- The model is necessarily a simplified representation of complex cellular dynamics and oscillator coupling. The authors should more clearly acknowledge where assumptions and approximations may limit accuracy.

- I would recommend the authors thoroughly proofread the manuscript to correct typos, grammatical errors, and awkward phrasing. For example, the references are not in parentheses in the text, which makes the reading cumbersome; there are many typos, and the discussion is very short.

- Table 2 does not provide the units for the parameters.

- The authors explain that lattice sites are denoted by bold variables i = (i1,i2), and then use “j”, which has not been defined.

- Having schematic of the model, explaining the Cellular Potts Model, Hamiltonian function, time evolution, etc, maybe as part of the Methods section, would give a more intuitive explanation for the model.

Suggestions and questions:
- I would reorder the results by placing Figure 3 before Figure 2. Indeed, the authors first show their conclusions (Figure 2), and then justify why each zone corresponds to each phase state using the three different metrics (Figure 3). Presenting these metrics as tools to make conclusions about the different zones would make the case stronger.

- When the cells reach one state, do they stay in this state or do they continue to cycle (example, red cells in Figure 4)? If they continue to cycle, could you look at the timing of each phase according to the values of J and K? For example, in the cases where there is synchronization, do they always stay for the same amount of “time” in the same phase?

- The authors cite a few in vivo examples of such behaviors (“auditory sensory epithelia of many species”) – could they elaborate on the similarities and differences between their results and the biological observations from the litterature?

- In the same line of thinking: the K parameter determines the strength of adhesion of one cell to another – what happens when the adhesion is reversible, versus not?

- In addition to the three metrics used, the authors could compare the “remain incoherent” behavior to the behavior expected by chance.

Experimental design

no comment

Validity of the findings

no comment

Additional comments

no comment

Reviewer 2 ·

Basic reporting

Some typo errors:

For example: in the introduction, "intense are of study" should be corrected to "intense area of study",
"sytems" should be corrected to "systems".

Experimental design

Model Assumptions and Parameters:
The paper presents a model that utilizes the Cellular Potts Model (CPM) and incorporates internal oscillators. However, the model's underlying assumptions are not explicitly stated. It's important to note that every model has inherent assumptions, and it's essential to clearly outline these assumptions to comprehend the model's applicability and limitations.
Also, please clarify all the parameters in each equation. For example, in equation (1), the significance and physical meaning of parameter delta, Area, Atarget should be described clearly followed by the equation.

Parameter Choices:
The paper discusses different phase states that are based on parameters J and K. However, the justification for choosing these specific parameter values or ranges is missing. It is crucial to provide a rationale for these choices, particularly if they have any biological or practical significance. For instance, why was J= +0.6333 defined as "like attracts like" instead of any other value of J? A more thorough exploration or theoretical explanation for these observations would enhance the paper's strength and credibility.

Model complexity:
The paper’s objective is to model oscillators by considering them as entities with extended boundaries. However, it may oversimplify other crucial aspects. The paper only identifies "synchronization and aggregation" as the two primary interactions. This raises the question of whether there are other interactions or factors in real-world scenarios that the model might be overlooking.

Validity of the findings

Comparative Analysis:
The paper presents a comparison of its findings with those of previous models, but it lacks a clear validation to determine which model is more accurate or representative of real-world phenomena. Conducting a comparative analysis, which could involve the use of real-world data or more complex simulations, would help to validate the effectiveness of the new model.

Long-term Behavior:
While the paper discusses the dynamics of synchronization over time, it's unclear how the model behaves in the long run. Does it stabilize? Are there any emergent behaviors or patterns over extended periods?

Additional comments

no comment

---

## Round 0.2 · accepted · Accept

Dear Drs. Una and Glimm:

Thanks for revising your manuscript based on the concerns raised by the reviewers. I now believe that your manuscript is suitable for publication. Congratulations! I look forward to seeing this work in print, and I anticipate it being an important resource for groups studying modelling synchronization and aggregation. Thanks again for choosing PeerJ to publish such important work.

Best,

-joe

Reviewer 2 ·

Basic reporting

no comment

Experimental design

no comment

Validity of the findings

no comment

Additional comments

The authors' responses look good to me, and overall the manuscript improved after their revisions.